# Optimized Xenograft Protocol for Chronic Lymphocytic Leukemia Results in High Engraftment Efficiency for All CLL Subgroups

**DOI:** 10.3390/ijms20246277

**Published:** 2019-12-12

**Authors:** Sarah Decker, Anabel Zwick, Shifa Khaja Saleem, Sandra Kissel, Andres Rettig, Konrad Aumann, Christine Dierks

**Affiliations:** 1Department of Medicine I, Medical Center-University of Freiburg, Faculty of Medicine, University of Freiburg, 79106 Freiburg, Germany; sarah.decker@uniklinik-freiburg.de (S.D.); sashifee1990@gmail.com (S.K.S.); sandra.kissel@uniklinik-freiburg.de (S.K.); andres.rettig@uniklinik-freiburg.de (A.R.); 2Institute of Virology, Saarland University, 66421 Homburg/Saar, Germany; anabel.zwick@uni-saarland.de; 3Department of Pathology, Medical Center-University of Freiburg, Faculty of Medicine, University of Freiburg, 79106 Freiburg, Germany; konrad.aumann@uniklinik-freiburg.de; 4BIOSS Centre for Biological Signaling Studies, University of Freiburg, 79106 Freiburg, Germany; 5Division of Endocrinology, Diabetes and Clinical Nutrition, University Hospital, 8091 Zurich, Switzerland; 6Division of Hematology, University Hospital Zürich and University of Zurich, 8091 Zurich, Switzerland

**Keywords:** CLL, xenograft, microenvironment

## Abstract

Preclinical drug development for human chronic lymphocytic leukemia (CLL) requires robust xenograft models recapitulating the entire spectrum of the disease, including all prognostic subgroups. Current CLL xenograft models are hampered by inefficient engraftment of good prognostic CLLs, overgrowth with co-transplanted T cells, and the need for allogeneic humanization or irradiation. Therefore, we aimed to establish an effective and reproducible xenograft protocol which allows engraftment of all CLL subtypes without the need of humanization or irradiation. Unmanipulated NOD.Cg-*Prkdc^scid^Il2rg^tm1Sug^/*JicTac (NOG) mice in contrast to C.Cg-*Rag2^tm1Fwa-/-^Il2rg^tm1Sug^/*JicTac (BRG) mice allowed engraftment of all tested CLL subgroups with 100% success rate, if CLL cells were fresh, injected simultaneously intra-peritoneally and intravenously, and co-transferred with low fractions of autologous T cells (2%–4%). CLL transplanted NOG mice (24 different patients) developed CLL pseudofollicles in the spleen, which increased over 4–6 weeks, and were then limited by the expanding autologous T cells. Ibrutinib treatment studies were performed to validate our model, and recapitulated treatment responses seen in patients. In conclusion, we developed an easy-to-use CLL xenograft protocol which allows reliable engraftment for all CLL subgroups without humanization or irradiation of mice. This protocol can be widely used to study CLL biology and to explore novel drug candidates.

## 1. Introduction

Chronic lymphocytic leukemia (CLL) is characterized by a progressive accumulation of functionally incompetent CD5+ B-lymphocytes in the peripheral blood (PB), bone marrow (BM), and lymphoid organs [1,2]. CLL cells can account for up to 99% of circulating peripheral blood mononuclear cells (PBMCs), and thereby decrease the number of normal hematopoietic cells [3,4]. The disease has a highly variable clinical course. Some patients die from the disease within a few months, whereas others live for 20 years or more even without treatment [5]. Several prognostic markers such as Rai and Binet staging systems, immunoglobulin V_H_ gene mutational status [6], ζ-associated protein 70 (ZAP70) expression [7,8], cytogenetic abnormalities [9], and gene mutations can be used to predict the survival outcome and the need for treatment of patients with CLL [10,11]. While del 17p or del 11q CLL samples are highly aggressive and resistant to many chemotherapeutic agents, del 13q14 or a normal karyotype are associated with a good prognosis. In vitro monocultures of CLL cells are limited, and can only be used for drug screens based on short term readouts [12]. Co-culture of CLL cells with nurse-like cells or BM-stroma derived cell lines can maintain CLL survival for several days [13,14], but none of the available in vitro culture systems allow long-term culture or proliferation of CLL cells, which limits the relevance of such models for testing novel drugs [15].

Transgenic mouse models which show key features of human CLL have been developed by several groups to study the disease. The Eµ-TCL1 tg mouse model recapitulates many features of an aggressive CLL with increasing numbers of CD5+ CD19+ lymphocytes expanding in spleen, PB and BM leading to death of the mice within 1–1.5 years [16]. The double transgenic BCL-2:Traf2DN mouse model mimics refractory CLL and mice develop increased B cell counts with severe splenomegaly and lymphadenopathy by 6 months of age [17]. The cluster-deleted DLEU2/miR15a/16-1 replicates the deletion of the chromosomal region 13q14 and mice develop a more indolent CLL phenotype [18]. However, these transgenic mouse models do not yet cover the entire spectrum of human CLL subtypes. Notably, CLL characterized by major chromosomal changes like trisomy 12, del 17p or del 11q have not yet been amenable to transgenic methods based on single-gene modifications.

There are also notable limitations in the use of xenograft models which either use established human CLL cell lines [19,20], or mainly late stage and poor-prognosis primary CLL samples [21]. Thus, in 1997 the first CLL xenograft mouse model was developed [22]. CLL cells were injected into the peritoneal cavity of lethally irradiated Balb/c mice reconstituted with BM of SCID mice (lack of B and T cells), allowing short-term engraftment of late-stage CLL samples within the peritoneum for up to 2 weeks. Early stage CLL samples (RAI 0-I) only showed T cell engraftment, but their short-term engraftment could be improved by in vivo depletion of co-injected T cells with an OKT3 antibody [23]. The protocol was significantly improved by Dürig et al. using sublethally irradiated nonobese diabetes (NOD)/SCID mice as recipients which have additionally low NK cell activity and lack circulating complement. These researchers combined intra-peritoneal (i.p.) and intra-venous (i.v.) injections of PBMCs from CLL patients, and achieved a highly reproducible splenic and peritoneal engraftment with mainly late stage and poor prognostic CLL samples that remained stable for 4–8 weeks [24]. Other groups, using chemotherapy-based conditioning regimens with busulfan, have achieved similar results [21,25,26]. Systematic analysis of prognostic factors demonstrated favorable engraftment for CLL samples with poor prognosis including unmutated IGV_H_ genes, CD38 expression and ZAP70 expression [26]. Further progress was made by the use of humanized mice. The Chiorazzi group used an allograft setting and humanized irradiated NSG mice with human cord blood-derived CD34+ cells, human mesenchymal stromal cells, as well as mature human antigen-presenting CD14+ and CD19+ cells followed by injection of CLL-PBMCs, allowing CLL engraftment over several weeks. They found a clear correlation between numbers of proliferating autologous T cells with proliferation and frequency of engrafting CLL cells. In contrast to the previously published papers they showed that T cell proliferation is essentially required for CLL cell proliferation and that depletion of T cells completely blocked CLL engraftment in vivo [27]. Therefore, they further refined this protocol by activating CD3+ cells from CLL patients for 3–14 days with CD3/CD28 Dynabeads and IL-2 and transferred them with CLL-PBMCs into NSG mice via the retro-orbital vein [28,29,30]. Although they could engraft even good prognostic CLLs [27], major drawbacks of these models are the use of conditioning regimens, such as chemotherapy or irradiation, which irreversibly alter the microenvironment and the leukemic niche, and their reliance on allogeneic settings triggering immunological processes with unforeseeable implications on CLL biology. 

Therefore, the aim of the presented study was to establish an easy-to-use, effective, and reliable protocol to generate CLL xenograft mouse models with both poor and good prognosis CLLs, avoiding irradiation, chemotherapeutic conditioning regimens, or allogeneic settings. Specifically, we compared the CLL engraftment capacity of C.Cg-*Rag2^tm1Fwa-/-^Il2rg^tm1Sug^/*JicTac (BRG) versus NOD.Cg-*Prkdc^scid^Il2rg^tm1Sug^/*JicTac (NOG) mice in the absence of any conditioning regimen, as these mouse strains are the backbone for most future developments. Furthermore, we aimed to determine the role of fresh versus frozen samples, frequencies of co-transplanted T cells, and to validate the resulting optimized protocol with regard to treatment responses to the BTK inhibitor ibrutinib. Taken together, we established an easy-to-use protocol for xenografting all CLL subtypes which can be broadly used to validate novel treatment strategies and to study CLL biology in vivo. 

## 2. Results

### 2.1. Reliable CLL Engraftment is Detectable in NOG, but Not BRG Mice

Previous publications have shown huge advantages in CLL cell engraftment for late stage CLLs (Rai stage III/IV) [23,24,31] and CLL subsets with poor prognosis [21], but at least half of the patients seen in the clinic present with genetic and expression features of good prognosis and early stage CLL. Furthermore, many xenograft models rely on complicated conditioning regimens (chemotherapy, irradiation) or allogeneic settings. The aim of our study was therefore to establish an easy-to-use xenograft protocol for the good prognosis and early stage CLL subgroup, without any further interventions. NOG mice [32] and BRG mice [33] are currently the most immunocompromised basic mouse strains available, and are the backbone for the development of many new mouse models like humanized mice. CLL samples from 24 different patients were injected i.v. and i.p. into NOG and BRG mice and their engraftment capacity in the two different unconditioned mouse strains was compared. 

Figure 1A shows the engraftment of seven patients with good/intermediate prognosis and Rai 0-II into NOG versus BRG mice (Patient information Appendix A, Data analysis Appendix A). For each CLL sample tested, a total of 70 x 10^6^ freshly isolated CLL-PBMCs were injected into 9–11 weeks old recipients (*n* = 4/strain and Patient). As previously shown from the Dürig group, we observed the highest transplantation efficiency with combined i.v./i.p. injections of CLL PBMCs [24]. Four weeks after transplantation we could detect reproducible and stable engraftment (flow cytometry) of all the different CLL patient samples in NOG mice (Appendix A). Human cell engraftment detected by human CD45+ cells was highest in the spleen, followed by blood and BM. NOG mice demonstrated an enormous engraftment advantage over BRG mice in all analyzed patient samples (Appendix A, 4 mice per patient and mouse strain). 

Human cell recovery in NOG versus BRG mice 4 weeks after injection was 41.4% (NOG) versus 7.2% (BRG), huCD45+ cells in the spleen 29.8% (NOG) vs. 1.5% (BRG), and 8.8% vs. 0.2% in the BM. Total and relative numbers of CLL cells were significantly and markedly (>10-fold) higher in NOG mice compared to BRG mice, as shown in the spleen (35.604 vs. 1.167 CLL cells), femur (1.787 vs. 577 CLL cells), and blood (1.861 vs. 135 CLL cells) (Figure 1A, B; single data analysis Appendix A). Human cell populations were separated by using human CD45, CD5 and CD19 staining. CLL cells were CD5+CD19+CD45+, B cells CD5-CD19+CD45+ and T cells CD5+CD19-CD45+. Figure 1C demonstrates the gating strategy and shows a representative example for the engraftment of human CLL-, B- and T cells in NOG (left) versus BRG (right) mice, demonstrating the enormous differences in engraftment ability in the two different unmanipulated mouse strains.

### 2.2. Human CLL Cells and T Cells Expand during the First Weeks of Engraftment in NOG Mice, but Not in BRG Mice

To follow the course of human cell engraftment and human cell expansion over time, PB samples of CLL-PBMC transplanted NOG and BRG mice (4 mice per patient and mouse strain) as well as age-matched non-transplanted control mice (*n* = 3 per mouse strain) were collected two, four, and eight weeks after transplantation. In BRG mice, no increase in circulating human CLL cells, B cells or T cells was observed from day 14 to day 28. 

By contrast, there was a strong and significant increase in all circulating human cell types in the initial period after transplantation in NOG mice (day 14 to 28), indicating stable engraftment and even expansion of the human lymphoid cell compartments in this mouse strain. For instance, CLL cell numbers in the PB of NOG mice increased about 3-fold within 14 days (day 14 to 28) (Figure 2A). The expansion of human lymphoid cells in NOG mice was accompanied by progressive leukocytosis, anemia with reduced red blood cell (RBC), hemoglobin (HGB) and hematocrit (HCT) values, and pronounced thrombocytopenia at 28 days post-transplantation (Figure 2B). There was a positive correlation between engraftment of CLL cells in BM and PB, while spleen engraftment was completely independent (Figure 2C).

At 8 weeks after transplantation, the number of CLL- and B- cells slowly decreased, but was still well detectable. Numbers of human T cells stayed at equal levels between d28 and d56 (Appendix A).

### 2.3. CLL Cells Rebuild Lymphoid Follicles in the Spleen of NOG Mice, but Not BRG Mice 

Next, we compared the engraftment patterns of human CLL cells in spleen and liver tissues of NOG and BRG recipient mice. At 28 days post-transplantation, we observed increased spleen and liver weights in the CLL-PBMC transplanted mice compared to age-matched, non-transplanted control mice (Figure 3A). While BRG mice showed only minor, non-significant spleen weight increases, the spleen weights in NOG mice more than doubled (mean values: 0.025 g compared to 0.089 g) as a result of CLL transplantation (Figure 3A). Next, spleen and liver sections were stained with NACE, to identify remaining murine granulocytes, and for human CD45, CD20 and CD7, to identify transplanted CLL and T cells. In the non-transplanted control setting, both NOG and BRG mice are characterized by small spleen sizes and lack of lymphoid follicles, with predominance of erythroid cells, granulocyte lineage cells and megakaryocytes (Figure 3B). By contrast, NOG mice at 28 days after injection of CLL cells show numerous nodular structures of lymphoid infiltrates displacing the erythroid and granulocytic compartments in the spleen (Figure 3B). 

Human CD20 staining demonstrated that the lymphoid follicle-like structures are repopulated with human CD45+/CD20+ cells, consistent with engraftment and nodular infiltration pattern of human CLL cells in these mice (Figure 3C + Appendix A). CLL follicle-like structures developed with all 24 tested CLL samples, including those with good prognosis (del 13q14, normal karyotype) and intermediate prognosis (trisomy 12) (Figure 3D). Furthermore, we could detect lambda or kappa light chains dependent on the CLL phenotype (Figure 3E + Appendix A), and found, as previously described, differentiation of CLL cells into plasma cells in about 10% of the cells within the lymphoid follicle in the spleens. In contrast, in BRG mice we could not detect any huCD20+ cells within the spleen or other tissues, and found a phenotypically untouched spleen in those mice (Figure 3C). 

In the NOG mouse transplantation model, CLL samples obtained from patients while receiving chlorambucil treatment also showed robust, and only slightly decreased, engraftment compared to their matched pretreatment control samples (Appendix A). In contrast, the transplantation of previously frozen CLL samples (*n* = 12) did not result in any engraftment >1% in NOG mice, which indicates that it is essential to use freshly isolated CLL cells for engrafting non-conditioned NOG mice.

### 2.4. Phenotypic Time Course of CLL Engraftment

White blood cell (WBC) counts, spleen weights and liver weights increased over the first 4 weeks (d 28), and then were stable until 8 weeks after CLL transplantation (Figure 4A,B). Immunohistochemistry (IHC) stainings revealed that during the first 4 weeks, splenomegaly was mainly induced by expanding CLL cells, which showed nodular CLL infiltrates reminiscent to CLL pseudofollicles in humans. Starting d 28 after transplantation, T cells started to surround the CLL cells (CD7 staining), and seemed to limit their further expansion, while the T cells themselves slowly infiltrated the complete spleen (Figure 4C + Appendix A).

### 2.5. Effect of Autologous T Cell Quantity on CLL Cell Engraftment in NOG Mice

Next, we aimed to understand whether we could completely deplete the human donor T cells from CLL grafts prior to transplantation without compromising engraftment efficiency. There are conflicting reports in the literature regarding the role of autologous T cells in CLL-xenograft transplantations, with some suggesting that they may be indispensable for engraftment [27] and others suggesting a benefit of their total depletion [23]. Based on our own set of experiments, we compared CLL cell engraftment in the transplantation with the highest number of co-engrafted T cells (T high, Tx2) with the CLL cell engraftment in the transplantation with the lowest T cell co-engraftment (T low, Tx4). Although the numbers of injected CLL-PBMCs were equal in both experiments, there was a more than 100-fold difference in engrafting T cell numbers 28 days after transplantation in the spleen, BM and PB among those two different sets of transplantations (Figure 5A). In contrast to T cell numbers, engrafting CLL cell numbers in the spleen were nearly equal 28 days after transplantation, regardless of initially transferred T cell numbers and regardless of the frequency of engrafted T cells at d28. CLL cell engraftment into the BM was better, if initial T cell co-transfers were low (Figure 5A). 

Next, we systematically compared CLL cell engraftment in spleens, BM and PB with T cell engraftment in the same organs. Further, we compared the T cell numbers and T cell fractions within the original graft in all performed experiments and for each mouse (Appendix A). While we did not find any clear correlation between T cell transfer numbers and CLL cell engraftment, we found that engrafting T cell numbers in the PB positively correlated with CLL cell engraftment in the PB (Appendix A). There was also a significant, albeit much lower, positive correlation between engrafting T cells in the spleen and engrafting CLL cells in the spleen (Appendix A), which indicates that T cells might at least in some contexts support CLL engraftment. In order to directly compare the influence of initial T cell co-transfers on CLL cell engraftment within the same CLL sample in our model, we adjusted T cell numbers experimentally by performing CD3 Macs separation to either partially reduce or deplete T cells. Partial T cell reduction from about 4.85% to 3.67% (Patient #11) and 8.87% to 3.40% (Patient #13) increased the total and relative numbers of engrafting CLL cells in all organs 28 days after transplantation 3–8 fold (Figure 5B,C). In contrast, near-total T cell depletion of 4.85% to 0.06% T cells (Patient #11) and 8.87% to 0.37% T cells (Patient #13) did not improve CLL engraftment (Figure 5B,C). Thus, our results confirm previous experiments from other groups and indicate that numbers of co-transferred T cells can indeed influence the engraftment of CLLs into NOG mice and frequencies of 2%–4% T-cells are required for good engraftment, while total depletion of T cells blocks sufficient CLL engraftment.

### 2.6. Model Validation Using Ibrutinib Treatment to Block CLL Expansion in Vivo

To test novel compounds for CLL patients, models are needed which closely recapitulate key features of the human disease. Despite the development of various CLL co-culture models, the long-term culture of primary CLL cells in vitro is challenging and only partially recapitulates treatment responses seen in CLL patients. Therefore, CLL xenograft models in mice hold promise in recapitulating further aspects of CLL biology, including the interplay with the microenvironment and the additional interaction with autologous transplanted T cells known to be clinically relevant in patients. Ibrutinib is highly effective in the treatment of CLL patients by blocking B-cell receptor signaling and microenvironmental interactions [34,35,36,37].

Previous xenograft mouse models have demonstrated efficacy of BTK inhibitors in vivo [25,38]. To investigate if our NOG/CLL xenograft mouse model is feasible for the validation of therapeutic agents, we treated CLL-PBMC transplanted NOG mice (*n* = 5 per group) with low-dose ibrutinib (25 mg/kg body weight qd p.o.) or vehicle control for 14 days, starting one week post-transplantation. During the course of treatment, ibrutinib caused a two-fold reduction in the number of CLL cells detected in the spleen and PB, but only a slight effect on CLL cell numbers in the BM (Figure 6A). Concomitantly, human B cell numbers were strongly reduced, while human T cell numbers remained unchanged, consistent with the BTK-kinase being selectively important for the B cell receptor bearing cell compartment. Immunohistochemistry confirmed that ibrutinib selectively and completely abrogated the observed nodular infiltration pattern of CLL cells in the spleen, with no apparent effects on infiltrating T cells (Figure 6D,E). These responses are remarkably similar to the clinical effects of ibrutinib in CLL patients. Notably, CLL patients first respond to ibrutinib treatment with reduced CLL cell infiltration in the spleen and in lymph nodes, but an increase in circulating CLL numbers in the PB due to enhanced mobilization of the CLL cells. A reduction in the number of CLL cells in the PB and also the BM begins to manifest itself only at later time points, up to several months after the start of ibrutinib treatment, implicating that our mouse model indeed closely recapitulates treatment responses seen in patients.

## 3. Discussion

To study CLL biology and to test novel compounds for treating CLL patients, models are needed which closely recapitulate key features of the human disease. Despite the development of various CLL co-culture models, the long-term culture of primary CLL cells in vitro is challenging and drug treatments only partially recapitulate treatment responses seen in CLL patients. Therefore, CLL xenograft models in mice hold promise to recapitulate further aspects of CLL biology, including the interplay with the microenvironment [14] and the additional interaction with autologous transplanted T cells known to be clinically relevant in patients [39,40,41].

Most CLL xenograft mouse models so far use conditioning regimens like irradiation or chemotherapy to prime the mice before CLL injection, with the consequence of destroying or at least changing the hematopoietic stem cell niche and the microenvironment in primary and secondary lymphoid organs [24,31]. Furthermore allograft settings were introduced to improve the CLL engraftment by adding allogeneic CD34+ cord blood cells and antigen presenting allogeneic CD14+ monocytes or CD19+ B cells, with the consequence of unforeseeable immunological changes in CLL biology [27]. To circumvent such implications on the microenvironment and the immune system, we intended to develop a CLL xenograft protocol without any conditioning regimens. Furthermore, we aimed to focus on early stage, and good prognosis CLL samples, as they are underrepresented in the previous studies despite their frequent occurrence [24,31].

In the initial experiments, we directly compared the engraftment efficiency of good/intermediate prognosis, and early stage (Rai 0–II) CLL cells into un-manipulated BRG versus NOG mice. Our xenograft studies with both mouse strains clearly demonstrated a huge advantage of the NOG mouse strain over the BRG mouse strain. Notably, the CLL engraftment in the spleen was increased more than 20-fold in NOG versus BRG mice. While we found a mostly nodular CLL infiltration pattern in the spleen of NOG mice, they were not present in BRG mice. Moreover, normal human B cells and human T cells, co-transferred with the CLL cells, showed much better engraftment in NOG versus BRG mice. While both mouse strains are devoid of murine T-, B- and NK cells and have downregulated cytokine receptor signaling pathways like IL2, IL4, IL7, IL9, IL15 and IL21, NOG mice show additional alterations in macrophage functionality and the function of the innate immune system regarding complement activation. 

Besides good prognostic CLLs, we then extended our studies to CLL samples under chemotherapy treatment and also to more aggressive late stage/poor prognosis CLLs. We found CLL engraftment in NOG mice with all tested conditions, indicating that our CLL xenograft model is feasible for all CLL subtypes, and is even functional for CLLs under chemotherapy. Nevertheless, the level of engraftment strongly differed in between patients without a clear correlation towards CLL genetics or clinical course. The most striking difference in engraftment capacity was shown in between fresh and frozen CLL cells, with a huge advantage towards freshly isolated CLL cells.

In contrast to previous studies using the NOG mouse strain, we did not irradiate our mice, nor did we humanize our mice with allogeneic CD34+ cord blood cells or antigen presenting cells like allogeneic CD14+ monocytes or CD19+ B-lymphocytes. These experimental simplifications in the protocol allow a very easy, effective and reproducible approach to establishing xenograft models for different CLL subtypes. Beyond the simplified handling in our protocol, the resulting xenografts may, in several regards, be more physiological compared to the other models. Firstly, the humanization of mice with human CD34+ cord blood cells or human CD14+ monocytes sets up an “allogeneic” context for the transplanted patient-derived CLL cells, on top of the xenogeneic context already provided in the recipient mice. This design will surely cause immune responses like graft-versus-leukemia effects which do not take place in CLL patients and are more reminiscent of an allogeneic BM transplantation setting. Secondly, a number of studies have shown that irradiation of mice severely alters the composition of niche cells within the BM and in other organs [42,43,44]. In the case of CLL, the malignant cells are vitally dependent on the interaction with an appropriate microenvironment [13,14], and niche alterations may severely alter their engraftment and expansion capacity. Our xenograft mouse model using unmanipulated NOG mice does not require additional allogeneic human stem cell support or the addition of allogeneic human antigen presenting cells. The reason for this might be the lack of irradiation in our model, which implies that microenvironmental interactions, as well as niche interactions, and also the viability and myeloid differentiation potential of murine HSCs is not hampered, and does not have to be replaced by human cells. Furthermore, we do not need to add allogeneic antigen presenting cells, as we find robust engraftment of autologous B cells and monocytes in addition to the CLL cells in the NOG model, which are co-transferred upon PBMC injection (Appendix A). While these considerations speak in favor of our NOG CLL xenograft protocol, there are also some trade-offs to be considered. Most obviously, our mouse model is associated with an uncontrolled expansion of donor-derived human T cells after 6–8 weeks, which may hamper the analysis of CLL biology that requires a longer follow-up period, and future studies aim to block the T cell expansion in vivo by the use of CD3/4/8 antibodies.

Further improvements for CLL engraftment without irradiation might come from humanized mouse models which express physiological levels of human IL-6, and allow improved B cell reconstitution and function [45]. Furthermore, Kit-deficient NOG mice (NBSGW = NOD,B6.SCID IL2rγ^-/-^ Kit^W41/W41^) show improved huHSC engraftment with a bias towards the lymphoid lineage and might therefore be valuable for CLL engraftment [46,47]. 

The role of different levels of autologous transplanted T cells co-transferred with CLL cells was previously examined in various xenograft mouse models and conflicting conclusions have been drawn regarding the T cell requirements for CLL engraftment. Bagnara et al. showed a direct correlation between frequencies of autologous T cells in the PB of xenografted mice and the expansion of the leukemic CLL cells [27]. In contrast, Dürig and Shimoni have shown that T cells inhibit CLL engraftment at least for early stage CLL samples and T cell depletion improves the CLL engraftment [23,24]. In our hands, near-total depletion of T cells reduces the engraftment of CLL cells, but, similar to results from humanized mice [26], a moderate reduction of T cell co-transfer to about 2% to 4% appears to be of advantage for CLL engraftment. Furthermore, we found that T cell numbers in the PB and spleen positively correlated with CLL cell numbers in the same organs, which indeed implicates that T cells at a moderate frequency support CLL growth in mice. Obviously, our analysis is not yet detailed enough to narrow down the quantitative perspective or to allow qualitative conclusions applicable to all CLL subtypes, but supports the previous findings from humanized mice [26]. 

Finally, we have established initial experimental support to validate our model in a setting of ibrutinib treatment in CLL. Similar to what has been reported in CLL patients [34,36,37], ibrutinib treatment effectively reduced the organ infiltration of the spleen with CLL cells within only two weeks of treatment, and the observed nodular infiltration pattern of CLL cells completely disappeared upon ibrutinib treatment. In contrast, there was no strong and significant response seen in the BM and PB. This response phenotype exactly reflects the human situation, where ibrutinib treatment shrinks lymph nodes and spleens within a very short time frame, but frequently results in increased CLL counts in the PB during the first weeks, which slowly declines over many months. 

In conclusion, we have developed a reliable and easy-to-use xenograft protocol (details shown in Table 1) for all CLL subtypes including patient samples with good prognosis CLL without the bias of humanization and irradiation, which can be broadly used to study CLL biology and to perform drug treatments.

## 4. Materials and Methods 

### 4.1. CLL Patient Samples

This study was approved by the Institutional Review Board of the University Medical Centre Freiburg (Reference number 44/08). Peripheral blood samples were obtained with informed consent in accordance with the Helsinki Declaration of 1975 from 24 B-CLL patients who were either untreated, or off therapy for at least 6 months (Appendix A). CLL cases were characterized for IgV_H_ mutational status, disease stage according to Binet and Rai criteria and history of treatment. Genetic aberrations were analyzed by chromosomal and FISH analysis and copy number changes were verified by SNP arrays (patient characteristics are shown in Appendix A). PBMCs were separated by Ficoll gradient centrifugation and used freshly. Some samples were frozen in fetal calf serum (FCS) with 10% dimethyl sulfoxide (DMSO) to compare their engraftment with the fresh samples (*n* = 12).

### 4.2. Transplantation of BRG and NOG Mice

All animal experiments were approved by the Regierungspräsidium Freiburg and were performed in accordance with the US National Institutes of Health Statement of Compliance with Standards for Humane Care and Use of Laboratory Animals. BRG (Balb/cRag2^-/-^IL2rg^-/-^ /C.Cg-*Rag2^tm1Fwa-/-^Il2rg^tm1Sug^/*JicTac) and NOG (NOD/Shi-scid IL2rg^-/-^NOD.Cg-*Prkdc^scid^Il2rg^tm1Sug^/*JicTac) mice were obtained from Taconic, and handled under sterile conditions. Mice were transplanted at 9 to 11 weeks of age with PBMCs from primary human CLL samples, isolated within 2 h of venipuncture by Ficoll density centrifugation. Therefore the interlayer cells were collected, washed twice, counted and resuspended in HBSS. A total of 70 × 10^6^ PBMCs was suspended in 0.4 mL HBSS, and transplanted using retro-orbital (0.2 mL) and i.p. (0.2 mL) injection. Disease development was monitored by weight measurements twice a week, biweekly blood cell counts, and FACS staining of the PB with APC-labelled anti-human CD45 antibody, PE-labelled anti-human CD19 antibody and V450-labeled anti-human CD5 antibody (all BD). Two, four or eight weeks after transplantation, mice were sacrificed, and spleen as well as BM cells were harvested and filtered unless noted otherwise. Erythrocyte lysis was followed by assessing total cell counts, and FACS staining for human CD45, CD19, and CD5. Cells were analyzed with the CyanADP flow cytometer (Beckman Coulter) and flow cytometric data were analyzed with the FlowJo 7.6 software (TreeStar). Engraftment was defined as > 1% human cells within spleen, bone marrow or peripheral blood.

### 4.3. Histology and Immunohistochemistry (IHC)

Spleens and tibiae were fixed in 4% buffered formalin, and BM was decalcified in an EDTA solution. After paraffin-embedding, sections were de-paraffinized in xylene and graded alcohols as described elsewhere [48]. Hemotoxylin and Eosin (H&E) and chloracetate esterase staining followed standard protocols. After specific antigen retrieval in “low pH target retrieval solution” (Dako) for 30 min, immunohistochemical staining was performed using antibodies against human CD45 (IR751, clone 2B11+PD7/26), human CD20 (IR604, clone L26), and human CD7 (M7255, clone CBC.37, all from DAKO, ready to use). The EnVision FLEX System (Dako) was used for signal visualization. Sections were counterstained with hematoxylin (Dako) and mounted.

### 4.4. T Cell Reduction or Depletion

Mononuclear cells were isolated from whole blood samples of CLL patients by Ficoll gradient separation, followed by CD3+ selection with anti-CD3 magnetic microbeads (Miltenyi Biotec) and separation with Macs Columns and Separators (Miltenyi Biotech). Purity was determined by flow cytometry.

### 4.5. Ibrutinib Treatment

Treatment of CLL-PBMC transplanted NOG mice started one week after transplantation and lasted for two weeks. Specifically, mice were divided into two subgroups (vehicle control and ibrutinib at 25 mg/kg body weight), matched for CLL cell count in the PB, total blood cell counts and body weight. Ibrutinib was obtained from Janssen-Cilag. Ibrutinib powder was dissolved in H_2_O with 30% PEG300 and 1% Tween20 as well as 1% DMSO. Ibrutinib was applied according to body weight using oral gavage. Disease development was monitored by daily body weight measurements. After 2 weeks of treatment, mice were sacrificed, and spleen and BM cells were harvested and filtered. Erythrocyte lysis was followed by total cell counts and FACS staining for human CD5, human CD19 and human CD45 (BD). Harvested cells were analyzed with the CyanADP flow cytometer (Beckman Coulter) and flow cytometric data were analyzed with the FlowJo 7.6 software (TreeStar). 

### 4.6. Statistical Analysis

Data is represented as the mean +/− standard error of the mean (SEM). Comparisons between parameters were performed using a two-tailed, paired or unpaired Student’s *t*-test (Prism5, GraphPad Software, San Diego, CA, USA). For all analyses, *p* < 0.05 was considered statistically significant. Correlations were assessed with the Pearson correlation coefficient. 

## Figures and Tables

**Figure 1 ijms-20-06277-f001:**
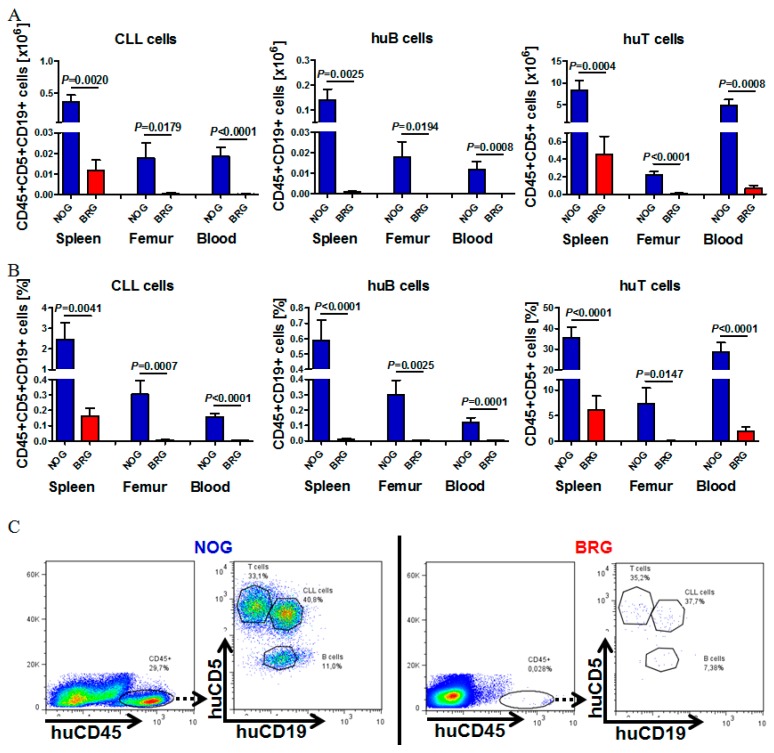
Engraftment comparison of chronic lymphocytic leukemia (CLL-), B- and T cells in spleen, bone marrow (BM) and peripheral blood (PB) of NOD.Cg-*Prkdc^scid^Il2rg^tm1Sug^/*JicTac (NOG) versus C.Cg-*Rag2^tm1Fwa-/-^Il2rg^tm1Sug^/*JicTac (BRG) mice. (**A**) Mean total human CLL-, B- (huB) and T cell numbers (huT) in the spleen, femur and PB of CLL-PBMC transplanted NOG and BRG mice 28 days after human cell injection (CLL patients #1–#7, *n* = 4 per mouse strain, total of 26–28 mice per mouse strain). (**B**) Mean relative numbers of human CLL-, B- and T cells engrafting in the spleen, BM and PB of NOG and BRG mice 28 days after transplantation (CLL patients #1–#7, *n* = 4 per mouse strain, total of 26–28 mice per mouse strain). (**C**) Representative example for flow cytometry analysis for huCD45, CD5 and CD19 of the spleens of NOG versus BRG mice four weeks after human cell injection (CLL patient #1). Human cell engraftment was analyzed by gating on human CD45+ cells, and the distribution of human CLL- (CD5+CD19+), B- (CD5-CD19+) and T cells (CD5+CD19-) by additional gating on CD5 and CD19.

**Figure 2 ijms-20-06277-f002:**
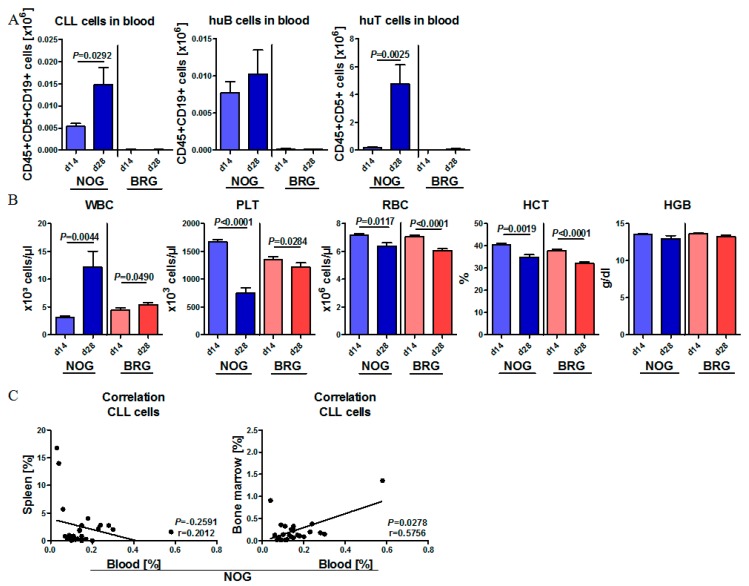
Human CLL cells and T cells expand during the first weeks of engraftment in NOG mice but not in BRG mice. (**A**) Mean human cell expansion in the PB of CLL-PBMC transplanted NOG (*n* = 26–28; 7 patients in 4 mice each) and BRG (*n* = 28; 7 patients in 4 mice each) mice two and four weeks after human cell injection (CLL patients #1–#7). CLL cells were characterized as human CD45+, CD19+ and CD5+; human B cells were detected as CD45+, CD5- and CD19+; human T cells were detected as huCD45+, CD19- and CD5+. (**B**) Mean white blood cell counts (WBC), red blood cell counts (RBC), and platelet (PLT) counts, hemoglobin and hematocrit of non-transplanted age-matched control mice, CLL-PBMC transplanted NOG (*n* = 26–28; 4 per 7 patients) and BRG mice (*n* = 28; 4 per 7 patients), analyzed at days 0, 14 and 28 after transplantation (CLL patients #1–#7). (**C**) Correlation of CLL cell engraftment in the blood with CLL cell engraftment in the spleen (left) or BM (right) of transplanted NOG mice four weeks after transplantation (CLL patients #1–#7) using Pearson correlation calculation.

**Figure 3 ijms-20-06277-f003:**
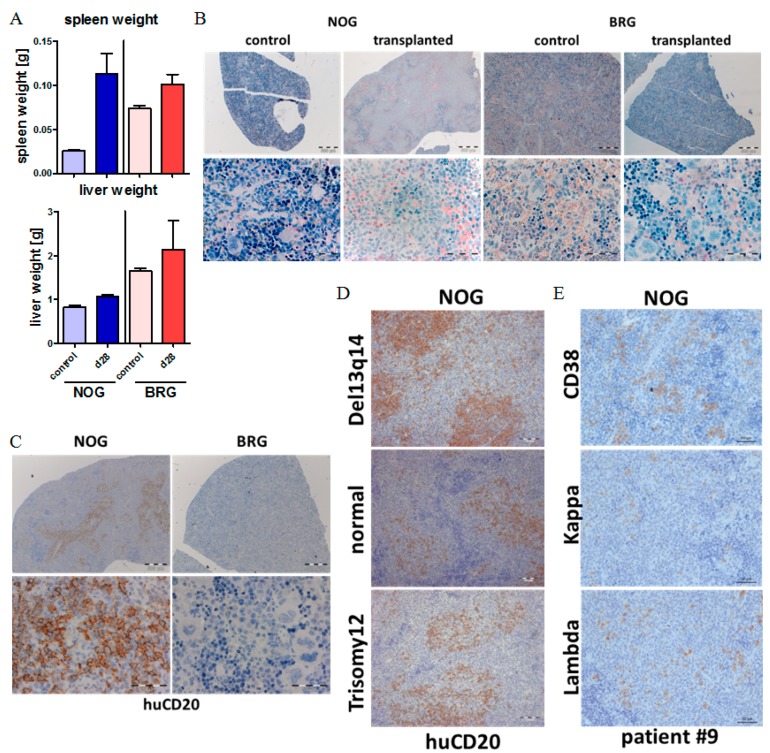
CLL cells rebuild lymphoid follicles in the spleen of NOG mice but not in BRG mice.(**A**) Mean spleen and liver weights of CLL-PBMC (peripheral blood mononuclear cells) transplanted NOG (*n* = 26, CLL Patients #1–#7)) and BRG (*n* = 28, CLL Patients #1–#7)) mice, and age-matched non-transplanted control mice at day 28 post transplantation. (**B**) NACE staining of spleen sections of transplanted NOG and BRG mice four weeks after human cell injection (CLL Patient #1). Image size: ×25 original magnification for upper images (bar indicates 500 µm) and ×400 original magnification for lower images (bar indicates 50µm). (**C**) Immunohistochemistry (IHC) analysis of spleen samples from CLL-PBMC transplanted NOG and BRG mice stained for human CD20, at day 28 after human cell injection (CLL Patient #1). Image size: ×25 original magnification for upper images (bar indicates 500 µm) and ×400 original magnification for lower images (bar indicates 50 µm). (**D**) IHC analysis of spleen samples from CLL-PBMC transplanted NOG mice stained for human CD20, at four weeks after human cell injection (CLL patients #3, #4 and #2). Image size: ×200 original magnification for images (bar indicates 50 µm). (**E**) IHC analysis of spleen samples from CLL-PBMC transplanted NOG mice stained for human CD38 and Kappa and Lambda light chain at day 28 after human cell injection (CLL patient #9). Image size: ×200 original magnification for images (bar indicates 50 µm).

**Figure 4 ijms-20-06277-f004:**
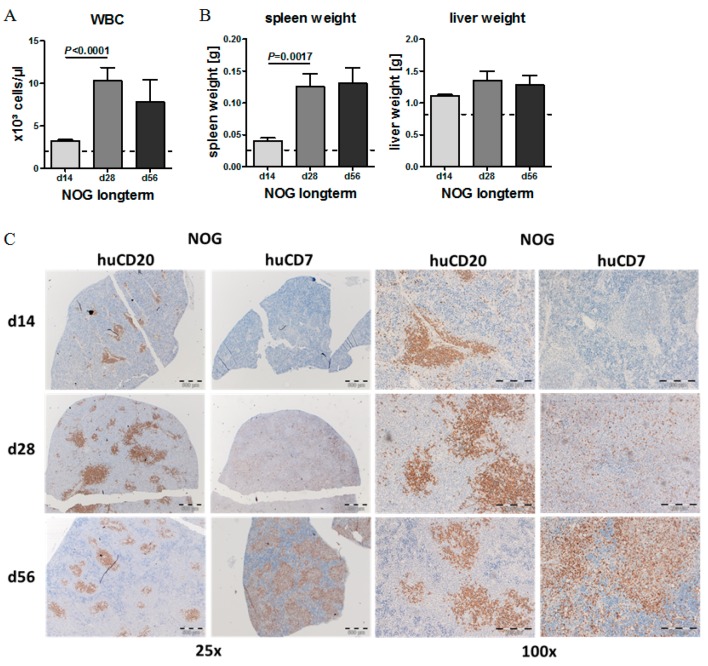
Long-term engraftment of human CLL cells in NOG mice. (**A**) Mean WBC counts in the PB of transplanted NOG mice 14, 28 and 56 days after human cell injection (CLL Patients #9–#10, *n* = 3 per patient and time point). (**B**) Mean spleen and liver weights of CLL-PBMC transplanted NOG mice at two, four and eight weeks after transplantation (CLL patients #9–#10, *n* = 3 per patient and time point). (**C**) IHC stainings of spleens from CLL-PBMC transplanted NOG mice for human CD20 (right) and CD7 (left) 14, 28 and 56 days after human cell injection (CLL Patient #9). HuCD20 shows B-CLL cells and CD7 human T cells. Image size: x25 original magnification for upper images (bar indicates 500 µm) and ×100 original magnification for lower images (bar indicates 200 µm).

**Figure 5 ijms-20-06277-f005:**
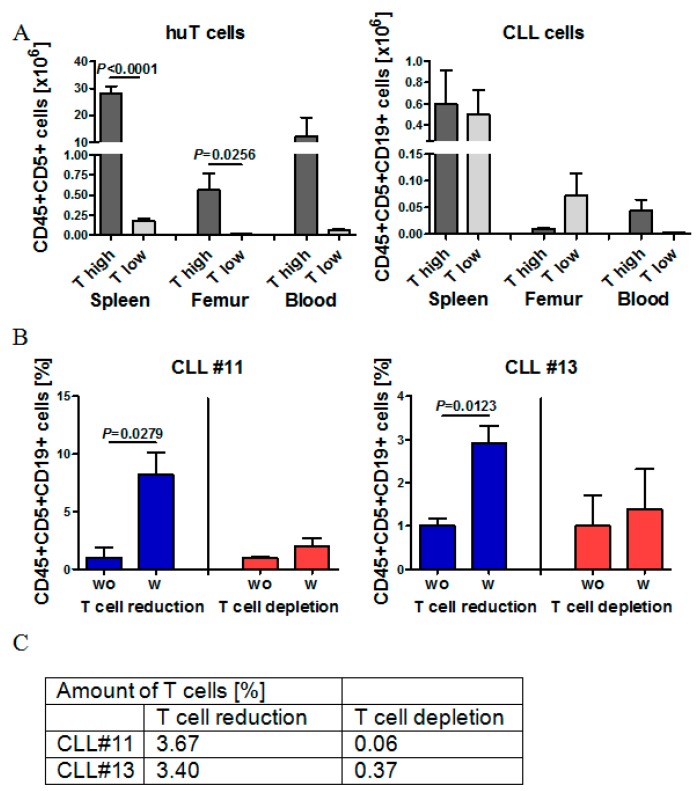
The role of autologous T cells on CLL cell engraftment in NOG mice. (**A**) Mean numbers of human T cells and CLL cells in the spleen, femur and PB of NOG mice of transplantation T high (CLL patient #2, *n* = 4) and T low (CLL patient #4, *n* = 3). (**B**) Direct comparison of relative CLL cell engraftment in NOG mice receiving either CLL-PBMCs (*n* = 3, wo) or CLL-PBMCs with reduced or depleted autologous T cell fractions (*n* = 3, w). Left graph shows CLL#11 and right graph CLL#13. (**C**) Table depicts the amount of human T cells in CLL-PBMCs of CLL#11 and CLL#13 after T cell reduction or T cell depletion before injection into NOG mice.

**Figure 6 ijms-20-06277-f006:**
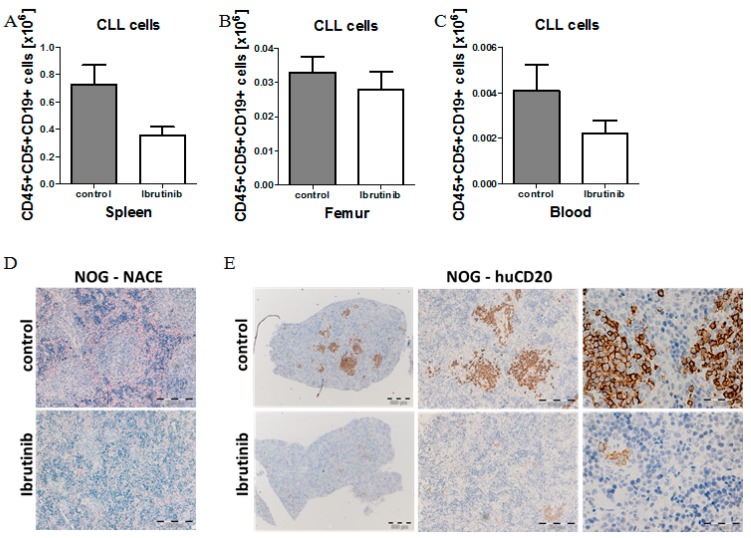
Model validation using ibrutinib treatment to block CLL infiltration in NOG mice. (**A**) Mean number of CLL cells in the spleen of CLL-PBMC transplanted NOG mice (CLL patient #12) with control vehicle treatment (*n* = 5) or after 14 days of ibrutinib (25 mg/kg BW) treatment (*n* = 5) which was started one week post-transplantation and was administered once a day orally. (**B**) Mean number of CLL cells in the femur of CLL-PBMC transplanted NOG mice (CLL patient #12) after two weeks of ibrutinib treatment (25 mg/kg BW, *n* = 5). (**C**) Mean number of CLL cells in the PB of CLL-PBMC transplanted NOG mice after 14 days of ibrutinib treatment (25 mg/kg BW, CLL patient #12, *n* = 5). (**D**) NACE staining from spleen sections of CLL-PBMC transplanted control and ibrutinib treated (25 mg/kg BW, CLL patient #12, *n* = 5) NOG mice. Image size: x100 original magnification for images (bar indicates 200 µM). (**E**) IHC staining for human CD20 in the spleen of CLL-PBMC transplanted NOG mice after two weeks of ibrutinib treatment (25 mg/kg BW, CLL patient #12, *n* = 5. Image size: x25 original magnification for left images (bar indicates 500 µm), ×100 original magnification for images in the middle (bar indicates 200 µm) and ×400 original magnification for right images (bar indicates 50 µm)).

**Table 1 ijms-20-06277-t001:** CLL xenograft protocol.

1. Take 20–40 mL of fresh peripheral blood (PB) in EDTA tubes from CLL patients (>20.000 WBC/µL)
2. Isolate PBMCs (CLL cells, T- and B cells, monocytes) by Ficoll density centrifugation
3. Count isolated PBMCs, quantify CLL cells, T cells and B cells by flow cytometry (CD19, CD5) Step 1–3 within 3 h
4. (a) T cell quantity <4% proceed to step 5 (b) T cell quantity >4%: MACS-separation with CD3+ selection
5. Dilute 70 x 10 E6 CLL-PBMCs in 400 µL of HBSS
6. Inject cell suspension half i.p. and half retro-orbital into 8–11 weeks old NOG mice
7. For treatment studies: Start at day 7 post transplantation, 14–21 days of treatment
8. Final analysis: (a) Organ weights: spleen and liver (b) Histology: liver, 1/3 of spleen and 1 tibia (c) Cell counts: 2/3 of spleen and 1 femur (d) FACS: stain cells from spleen, BM and PB for human CD45, CD19 and CD5

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
