# Peer review of "Optimized Xenograft Protocol for Chronic Lymphocytic Leukemia Results in High Engraftment Efficiency for All CLL Subgroups"

_ijms, 2019, doi:10.3390/ijms20246277_

Round 1

Reviewer 1 Report

Authors established xenograft protocol of CLL without humanization and irradiation. This manuscript is well-written and there are no points to be revised.

Author Response

Dear reviewer 1

Thank you for your comment and the positive statement.

Reviewer 2 Report

The authors well present their study of CLL engraftment in the NOG mice compared to other model BRG model. The data are well introduced and presented. Interesting are also data on the effect of T cells (at certain dose) on the CLL engraftment. Interesting was also treatment of egrafted CLL with Ibrutinib, although the significance of differences and effects on survival (OS) were not presented. Together, this work I found acceprable. The paper might attempt also to answer or comment what is a general effect of the CLL engrafted in the NOG mice on the OS. It is not really required to do survival study but only in case the authors can comment on this or present some survival data e.g. for adverse prognostic CLL vs good prognosis?

Author Response

Dear reviewer 2

Within our paper we investigated the CLL engraftment capacity of NOG versus BRG mice upon transplantation with good and bad prognosis CLLs. We did not do survival studies comparing good- and bad prognostic CLLs in those two models, therefore we cannot comment on that. The general problem with survival studies in this xenograft model is the development of GVHD by the co-transplanted autologous T-cells. Therefore the survival of the mice is not only limited by the onset of CLL disease, but even more by the GVHD, and it is hardly possible to seperate those two life limiting events. Therefore most xenograft studies do not use overall survival as study endpoint.